

# 1 O₃ and PAN in southern Tibetan Plateau determined by distinct
# 2 physical and chemical processes

Wanyun Xu[1], Yuxuan Bian[1], Weili Lin[2], Yingjie Zhang[3,a], Yaru Wang[3,b], Gen Zhang[1,*], Chunxiang
Ye[3,*] , Xiaobin Xu[1]
[1] State Key Laboratory of Severe Weather & Key Laboratory for Atmospheric Chemistry of CMA, Institute of Atmospheric
Composition, Chinese Academy of Meteorological Sciences, Beijing, 100081, China
[2] College of Life and Environmental Sciences, Minzu University of China, Beijing, 100081, China
[3] College of Environment, Peking University of China, Beijing, 100871, China
[a] now at School of Ecology and Nature Conservation, Beijing Forestry University, Beijing, 100083, China
[b] now at Leibniz Institute for Tropospheric Research, 04318 Leipzig, Germany
*Correspondence to*: Gen Zhang (zhanggen@cma.gov.cn) and Chunxiang Ye (c.ye@pku.edu.cn)
**Abstract.** Tropospheric ozone ($O_3$) and peroxyacetyl nitrate (PAN) are both photochemical pollutants harmful to the
ecological environment and human health. In this study, measurements of $O_3$ and PAN as well as their precursors were
conducted from May to July 2019 at Nam Co station (NMC), a highly pristine high-altitude site in the southern Tibetan
Plateau (TP), to investigate how distinct transport processes and photochemistry contributed to their variations. Results
revealed that, despite highly similar diurnal variations with steep morning rises and flat daytime plateaus that were caused by
boundary layer development and downmixing of free tropospheric air, day to day variations in $O_3$ and PAN were in fact
controlled by distinct physiochemical processes. During the dry spring season, airmasses rich in $O_3$ were associated with
high altitude westerly airmasses that entered the TP from the west or the south, which frequently carried high loadings of
stratospheric $O_3$ to NMC. During the summer monsoon season, a northward shift of the subtropical jet stream shifted the
stratospheric downward entrainment pathway also to the north, leading to direct stratospheric $O_3$ entrainment into the
troposphere of the northern TP, which travelled southwards to NMC within low altitudes via northerly winds in front of
ridges or closed high pressures over the TP. Westerly and southerly airmasses, however, revealed low $O_3$ levels due to the
overall less stratospheric $O_3$ within the troposphere of low latitude regions. PAN, however, was only rich in westerly or
southerly airmasses that crossed over polluted regions such as Northern India, Nepal or Bangladesh before entering the TP
and arriving at NMC from the south during both spring and summer. Overall, the $O_3$ level at NMC was mostly determined
by stratosphere-troposphere exchange (STE), which explained 77% and 88% of the observed $O_3$ concentration in spring and
summer, respectively. However, only 0.1% of the springtime day-to-day $O_3$ variability could by STE processes, while 22%
was explained during summertime. Positive net photochemical formation was estimated for both $O_3$ and PAN based on
observation-constrained box modelling. Near surface photochemical formation could not explain the high $O_3$ level observed
at NMC and was also not the factor determining the day-to-day variability of $O_3$, however, it captured events with elevated
PAN concentrations and was able to explain its diurnal variations. Both $O_3$ and PAN formation were highly sensitive to $NO_x$
levels, with PAN being also quite sensitive to VOCs concentrations. Under the rapid development of transportation network



and the urbanization inside the TP, increased emissions and loadings in $NO_x$ and VOCs might lead to strongly enhanced $O_3$
and PAN formation in downwind pristine regions, which should be paid more attention in the future.

## 1 Introduction

Ozone ($O_3$) and peroxyacetyl nitrate (PAN) are both key photochemical pollutants within the troposphere, that are harmful to
vegetation and human health (Kleindienst et al., 1990;Yukihiro et al., 2012;Taylor, 1969;Lefohn et al., 2017). Since $O_3$ and
PAN are both produced during the oxidation of volatile organic compounds (VOCs) in the presence of nitrogen oxides ($NO_x$),
they often share highly similar variational characteristics (Fischer et al., 2014). However, PAN is formed only from a limited
number of oxygenated VOCs (OVOCs), which are typically oxidation products of alkenes (with low carbon numbers),
aromatics and isoprene (Xu et al., 2021), while $O_3$ can be practically formed from all VOCs. Additionally, the photochemical
formation of $O_3$ depends highly nonlinearly on its precursor concentrations, being insensitive to VOCs changes under $NO_x$-
limited conditions and vice versa, while PAN varied nearly proportional to its OVOCs precursors, with additional influences
from the $NO_2$ to NO ratio (Xu et al., 2021). Thus, photochemistry can sometimes result in distinct variations of $O_3$ and PAN,
especially during cold seasons (Xu et al., 2021;Zhang et al., 2020). From the aspect of physical transport, $O_3$ and PAN can be
both transported over large distances. Since PAN is easily thermal decomposed under high temperatures, its transport is
more favored in cold seasons or at higher altitudes. Early simulation studies suggested PAN to be an important reservoir for
$NO_x$ in the troposphere and lower stratosphere (Singh and Hanst, 1981) and redistributes $NO_x$ far from its source regions
(Moxim et al., 1996). Different from PAN, $O_3$ is naturally produced within the stratosphere and can be transported into the
troposphere via stratosphere-troposphere exchange (STE) processes that are often associated with the occurrence of
tropopause folds, cut-off lows, streamers near the polar-front jet and subtropical jet stream and mid-latitude cyclones
(Langford, 1999;Stohl et al., 2003;Sprenger et al., 2007;Tang et al., 2011). STE elevates tropospheric $O_3$ and oxidation
capacity at distinct latitudes during different seasons, with overall largest mass fluxes in summer occurring mostly in higher
mid latitudes, followed by spring occurring mostly in lower mid latitudes (Tang et al., 2011;Škerlak et al., 2014). Deep STE
intrusions reaching the planetary boundary layer (PBL) and associated mass fluxes are largest during spring in China and the
western part of North America.
While the variational characteristics, influence of photochemical formation and transport on $O_3$ and PAN have been widely
investigated in polluted urban regions of China (Liu et al., 2018;Yao et al., 2019;Hu et al., 2020;Wei et al., 2020;Qiu et al.,
2021;Zhang et al., 2021;Xu et al., 2021), those at remote background sites received less attention, especially for PAN. The
Tibetan Plateau (TP), located in western China, is often called the "Third Pole" with its average altitude over 4000 m. Due to
its harsh environment, the TP is only scarcely populated and thus highly pristine. The topography of the TP affects large
scale circulations with its strong thermal forcing, thereby influencing the weather, climate and air quality in eastern China
(Yang et al., 2014). Surface $O_3$ as a crucial greenhouse gas and with its deterministic role on atmospheric oxidation capacity



has been paid certain attention in the TP and special concern has been paid to the photochemical formation of $O_3$ under the
strong radiative conditions at such high altitudes. Ma et al. (2002) investigated the photochemical formation of $O_3$ at Mt.
Waliguan (WLG) in the Northeastern TP through box-modelling and suggested wintertime net production and summertime
net loss in $O_3$. Xue et al. (2013) further constrained the box model with VOCs sampling results, which mainly included
hydrocarbons and aromatic compounds (no oxygenated compounds), and found net $O_3$ formation at WLG during both spring
and summer 2003. Transport from central and eastern China was found to be frequent during summertime (Xue et al.,
2011;Xu et al., 2018a), which revealed higher $O_3$ production efficiencies (Xue et al., 2011) and was responsible for rising $O_3$
trends during summer and autumn (Xu et al., 2016;Xu et al., 2018a). Due to its high altitude, the TP revealed the largest deep
STE $O_3$ mass fluxes, with higher fluxes in spring and winter and lower ones in summer and autumn, especially in the
southeastern TP (Škerlak et al., 2014). At WLG, $O_3$ was observed to be strongly influenced by STE associated with the
subtropical jet during spring and summer in 2003, with stronger impacts during summer than spring (Ding and Wang,
2006;Zheng et al., 2008). STE was estimated to contribute an annual average of 10.2% to tropospheric $O_3$ at WLG based on
EMAC model simulations using tagged tracers, revealing a peak contribution in June (Liu et al., 2020). At Nam Co station
(NMC) in the southern TP stratospheric influence was also mainly observed during spring and summer, which was estimated
to contribute 20% and 10%, respectively, based on model simulations (Yin et al., 2017). Measurements from Dangxiong, a
lower site not far from NMC, also revealed significant stratospheric impacts on surface $O_3$ (Lin et al., 2015). At Xianggelila
station in the southeastern TP, the STE impact was suggested to be most pronounced during winter and weakest during
spring and summer based on surface observations (Ma et al., 2014), which however was in disagreement with modelling
results revealing strongest STE during April and May, with an annual average contribution of 4.3%. In comparison, PAN
was far less investigated, the few existing studies mainly focused on the impact of transport on local PAN variations. Zhang
et al. (2009) made measurements of $O_3$ and PAN at WLG station during summer 2006 and found that the two oxidants
exhibited distinct diurnal variations and only weak correlations to each other, suggesting they were controlled by different
processes, with PAN being strongly influenced by regional transport of polluted air plumes. Xue et al. (2011) analyzed the
same set of observations and reported PAN to be one of the most abundant reactive nitrogen species ($NO_y$) at WLG,
contributing 32% to total $NO_y$. Xu et al. (2018b) made measurements of $O_3$ and PAN at NMC station in summer 2011 and
late spring to early summer 2012, detecting highly similar diurnal variations in both gases caused by boundary layer
development and elevated PAN in connection with transport of air plumes crossing over Nepal, North Pakistan or North
India.
Despite the findings in previous literature, the physiochemical factors determining the variation of $O_3$ and PAN in the TP and
their relative contributions have not been comprehensively investigated mainly due to the lack of comprehensive online
VOCs observations and accurate $NO_x$ measurements. In this study, we present integrated real-time measurements of
$O_3$, PAN, $NO_2$, VOCs, $CH_4$, CO, photolysis rates and other meteorological parameters during spring and summer 2019 at
NMC station and analyzed them in combination with reanalysis data, utilizing trajectory modelling and box-modelling
approaches. The different impact of distinct transport processes and photochemical formation on $O_3$ and PAN, as well as





differences in sensitivities towards their precursors are intercompared using improved box-model constraints and the relative
contributions of physical and chemical processes to $O_3$ and PAN variability are evaluated.
**2 Experimental and analysis methods**
**2.1 Site, observations and data**
As the first part of the @Tibet series campaign, a campaign was carried out at NMC Station (30.77° N, 90. 95° E, 4730m
a.s.l.), which is a highly pristine site in the southern TP (Fig. 1). The site campus is located within the natural reserve of
NMC Lake, thus far away from anthropogenic activities and emissions. The nearest county (Dangxiong) and city (Lhasa) are
located 40 and 125 km to the southeast of NMC, respectively. The NMC Lake was ~1 km north to our observation site,
while the foothills of the northern Nyainqêntanglha Mountains were ~15 km to the south.
Measurements were performed from 1 May to 31 July 2019. Instruments for gases (including $O_3$, PAN, $NO_2$, CO, $CH_4$ and
non-methane volatile organic compounds (NMVOCs)) were housed in an air-conditioned container. $O_3$ was measured
alternatingly at the heights of 1.8 and 6.8 m (switching between two heights at 15-minute intervals) using a Model TE-49C
commercial $O_3$ analyzer, which was calibrated with a TE-49iPS $O_3$ calibrator (both from Thermo Electronics, USA). A Los
Gatos Research (LGR) $NO_2$ Analyzer was adopted for the measurements of $NO_2$, which has a measurement range of 0.01-
1000 ppb and was calibrated using $NO_2$ standard gas at the beginning and end of the experiment. PAN measurements were
made using a GC-ECD analyzer (Meteorologie Consult GmbH, Germany) which was calibrated using PAN instantly formed
in the reaction of a NO reference gas with acetone in the internal calibration unit of the instrument. CO and $CH_4$ were
measured (until 2 July) by a cavity ring-down spectroscopy (CRDS) analyzer (Model G2401, PICARRO, USA) at a high
precision (0.4 and 0.04 ppb, respectively, for CO and $CH_4$). The CRDS instrument was calibrated twice using a mixed CO
and $CH_4$ standard gas, which was pressurized in 29.5 L treated aluminum alloy cylinders (Scott-Marrin Inc.) fitted with high-
purity, two-stage gas regulators, and calibrated with cylinders assigned by the Global Atmosphere Watch (GAW) CO
Central Calibration Laboratory operated by National Oceanic and Atmospheric Administration (NOAA) Earth System
Research Laboratory (ESRL). NMVOCs were measured (only from 29 April to 21 May) using an online GC-MS/FID
analysis system (TH-PKU 300B, Wuhan Tianhong Instrument Co. Ltd., China) at a 1-hour time resolution, with detection
limits in the range of 0.004 to 0.066 ppb.   Multipoint calibrations were performed using Photochemical Assessment
Monitoring Stations (PAMS) standard mixture and TO-15 standard mixture (100 ppb, Spectra Gases Inc., New Jersey, USA).
To account for the reactivity of different VOCs species, Propy-Equivalent VOCs concentrations were calculated as:
$C_{Propy-Equiv}(i) = C(i)\frac{k_{OH}(i)}{k_{OH}(C_3H_6)},$                     Eq. (1)
where C(i) is the ppbC concentration of species $i$ (calculated using ppb mixing ratios multiplied by carbon numbers of
species $i$), $k_{OH}(i)$ the reaction rate of species $i$ with OH radicals (obtained from master chemical mechanism,
http://mcm.york.ac.uk/MCM/), and $k_{OH}$ ($C_3H_6$) the reaction rate of propene with OH.





Photolysis rates (J values) were obtained using a Metcon CCD-spectrograph (Meteorologie consult GmbH, Germany),
whose receptor optics were mounted on top of the container at the height of 2 m. Conventional meteorological parameters
including temperature (T), relative humidity (RH), surface pressure (P), wind speed (WS) and wind direction (WD) were
recorded by an Automatic Weather Station. In addition, meteorological reanalysis data (ERA5) from the European Centre for
Medium-Range Weather Forecasts (ECMWF) were used for complimentary analysis.
**2.2 Backward trajectory analysis and PSCF calculations**
The HYSPLIT model (version 5) from NOAA Air Resources Laboratory (Draxler and Hess, 1997;Draxler and Hess,
1998;Draxler, 1999) was used for backward trajectory calculations, with 0.25° resolution GFS data from the National Center
for Environmental Prediction (NCEP) adopted as input. The trajectory endpoint was set at 250 m above the ground level of
NMC station. 7- day (168 hours) backward trajectories were calculated at an hourly interval for the entire period of the
campaign.
The potential sources of high $O_3$ and PAN were studied using the potential source contribution function (PSCF) analysis,
which has been widely applied to detect possible source regions (Ara Begum et al., 2005;Lucey et al., 2001;Zhou et al.,
2004). The PSCF on grid (i,j) is defined as:
$PSCF = m(i,j)/n(i,j),$ (1)
where m(i,j) is the residence time of a subset of trajectories, whereas n(i,j) is the residence time of all the trajectories in that
grid. Each trajectory was associated with $O_3$ and PAN concentrations observed at its time of arrival. To pin out the potential
source regions for high $O_3$ and PAN, the m(i,j) was calculated using the subset of trajectories that were associated with $O_3$ or
PAN concentrations higher than their respective 75[th] percentiles.
Abnormally high PSCF values may be produced for certain grids with very small n(i,j) values, which would induce large
uncertainties. Thus, a weighting factor $W(n_{ij})$ is introduced that was proposed by Zeng and Hopke (1989), giving grids with
few trajectories passing through less weight:
$$W(n_{ij}) = \begin{cases} 1.0, & n_{ij} > \overline{n_{ij}} \\ 0.7, & 0.1 \cdot \overline{n_{ij}} < n_{ij} \leq \overline{n_{ij}} \\ 0.4, & 0.05 \cdot \overline{n_{ij}} < n_{ij} \leq 0.1 \cdot \overline{n_{ij}} \\ 0.2, & n_{ij} \geq 0.05 \cdot \overline{n_{ij}} \end{cases},$$ (2)
where $\overline{n_{ij}}$ is the average number of $n_{ij}$.
The PSCF analysis was respectively performed for $O_3$ and PAN, separately for spring and summer periods. Based on
meteorological variations, spring and summer periods were defined as 1 May to 15 June and 15 June to 31 July, respectively.





**2.3 Box modelling of local photochemistry**

The Master Chemical Mechanism (version 3.2) was used within the F0AM (version 3.1) box-model framework developed by Wolfe et al. (2016), to simulate the impacts of local photochemistry on $O_3$ and PAN and to evaluate how much of their variations can be explained through local photochemistry. Observation data of VOCs, $NO_2$, J values and meteorological parameters were either averaged or interpolated into 10-minute averages and used as constraints in the model. Model simulations were only performed for the period from 1 to 21 May, when VOCs observation data were available. To evaluate local $O_3$ and PAN formation, three sets of simulations were performed for each, respectively using measurement constraints on OVOCs, $NO_2$ or both of them. In $O_3$ simulation cases, PAN was constrained by observations, while in PAN simulations $O_3$ was constrained. Daytime $O_3$ and PAN increments ($\Delta O_{3,mod}$ and $\Delta PAN_{mod}$) were calculated and compared against observed ones ($\Delta O_{3,obs}$ and $\Delta PAN_{obs}$), with their ratios used to reflect how much modelled local photochemistry can explain observed daytime increases in $O_3$ and PAN. Simulated $O_3$ and PAN net formation rates in distinct modelling scenarios were intercompared and to evaluated the sensitivity of their formation to VOCs and $NO_x$ concentrations.

**2.4 Impact of stratospheric-tropospheric exchange**

A Y index was defined as the ratio between normalized $O_3$ and water vapor concentrations, calculated using the following equation:

$$Y_{ind} = \frac{O_3/\overline{O_3}}{H_2O/\overline{H_2O}}. \qquad \text{Eq. (3)}$$

The Y index adopted in previous studies for the identification of stratospheric air intrusions has additionally divided Eq. (3) by normalized CO concentrations (Ma et al., 2014). Due to the lack of CO measurements after 2 July, the Y index was modified to the current form (in Eq. 3), which compared well with those calculated when incorporating normalized CO concentrations (Fig. S1), since CO revealed very small variability during the entire observation period.

Additionally, $O_3$ mass mixing ratios from the ERA5 hourly reanalysis dataset were converted to volume mixing ratios and applied in the investigation of STE impacts, since the ERA5 data are simulated with simple stratospheric $O_3$ chemistry consideration and thus mainly represents the physical transport of stratospheric $O_3$ (Sprenger and Wernli, 2003). Additionally, ERA5 $O_3$ data has been verified to be well representative of observed $O_3$ profiles and ground concentration levels at remote polar regions (Wang et al., 2021), indicating that it can well represent stratospheric $O_3$ and the influence of its transport.



## 3 Results and discussions

### 3.1 Variational characteristics of O3, PAN and their precursors

The time series of observed O$_3$, PAN, NO$_2$, CO, photolysis rates of O$_3$ (jO$^1$D) and NO$_2$ (jNO$_2$), as well as meteorological parameters observed at NMC from 1 May to 31 July are displayed in Figure 2. From 1 May to 15 June (defined hereafter as the spring period), NMC experienced cold temperatures, strong winds, and dry conditions with low relative humidity (RH) and hardly any precipitation except for three small snow events. While during 15 June to 31 July (defined hereafter as the summer period), temperatures increased, average wind speeds were smaller and frequent precipitation events occurred under the influence of the Asian summer monsoon (Fig. 2a, Table 1). Despite more frequent precipitation events, observed average daytime photolysis rates were similar between spring and summer periods.

Under such meteorological variations, O$_3$ and PAN exhibited higher average concentrations in the spring (59.8±13.4 and 0.27±0.08 ppb) and lower ones in the summer period (53.6±13.2 and 0.20±0.05 ppb), with O$_3$ levels being overall in accordance with previous observations (Xu et al., 2018b;Yin et al., 2017), while PAN levels were significantly lower than those observed in 2012 (Xu et al., 2018b). VOCs concentrations were only obtained for the first half of the spring period, reaching average concentrations of 4.1±3.5 Propy-Equiv. ppbC, to which OVOCs contributed 61±12% (reaching 2.5±2.2 Propy-Equiv. ppbC on average), followed by alkenes (0.6±0.6 Propy-Equiv. ppbC), aromatics (0.6±1.3 Propy-Equiv. ppbC) and alkanes (0.47±0.5 Propy-Equiv. ppbC), which made up similar fractions (14±6%, 13±7%, and 11±4%, respectively), while other components (including alkynes, halogenated VOCs and nitriles) had negligible impacts (1±1%) on the overall VOC concentration and reactivity (Fig. S2). While daytime concentrations of OVOCs and alkenes were significantly higher than those during nighttime, other VOCs species did not display much day-night discrepancy. NO$_2$ revealed averaged concentrations 0.12±0.05 and 0.09±0.05 ppb during spring and summer periods, respectively, with no evident day to night differences.

Averaged diurnal variations of O$_3$ and PAN resembled each other (Figs. 3a1-2), both revealing decreases after sunset, reaching lowest values at 7:00 Beijing Local Time (LT) and increasing quickly after sunrise simultaneous to PBL height (PBLH, Figs. 3b1-2) and exhibiting a flat plateau afterwards. Both O$_3$ and PAN revealed higher levels in spring and lower ones in summer, however, PAN concentrations have decreased more significantly than O$_3$ (26% versus 10%), revealing a very flat and broad plateau during the day. OVOCs, aromatics and alkenes determined the variations of VOCs, with OVOCs and alkenes displaying diel variations similar to those of O$_3$ and PAN, revealing increases from 7:00 to 9:00 LT, fluctuating around its daily maximum value over daytime and decreasing after 17:00 LT (Figs. 3b1-2). NO$_2$ was typically higher during nighttime and lower during daytime, which is caused by combined effects of weakened dilution under nighttime shallow boundary layers, natural and anthropogenic NO$_x$ emissions, as well as chemical transformations. Additionally, springtime NO$_2$ concentrations were higher than those during the summer period (Figs. 3c1-2). CO, however, revealed only slightly higher concentrations during the summer period, staying overall flat during the day, without any diurnal variations (Figs. 3c1-2). Both RH and absolute water vapor concentrations were higher during the summer period. RH revealed a diurnal



maximum by 7:00 LT during both periods, decreased rapidly after sunrise and reached its diurnal minimum at 16:00 and
18:00 LT in spring and summer, respectively (Fig. 3e1-2). Water vapor, however, increased further after sunrise, possibly
due to surface evaporation processes of frost and dew during the morning. While the diurnal peak in photolysis rates were
similar between spring and summer periods, the averaged diurnal variations displayed a narrower peak during summer,
especially for $jNO_2$, due to more frequent precipitation and higher cloud coverage.
The day-to-day evolution of diurnal $O_3$ and PAN variations as well as those of winds and PBLH are more clearly displayed
by Figs. 4). Downward winds were strongest during the afternoon under high PBLH (Figs. 4a,d). Due to the local
topography with the NMC Lake to its west and north and the Nyainqêntanglha Mountains to its south, the site was
susceptible to both influences from land-lake and mountain-valley breezes. Accordingly, local surface winds displayed clear
diurnal variations with southeasterly nighttime winds shifting to northwesterly winds during daytime (Figs. S3b-d, Fig. S3 is
the same as Fig. 4, with winds replaced by 2 m wind measurements). 550 hPa winds from ERA5 over the 0.25° grid
containing NMC station (representing near surface conditions, since surface pressure was on average 573±2 hPa) revealed
stronger diurnal variations in zonal winds (Fig. 4b), overall agreeing with variations in surface winds, while meridional
winds were dominated by southerly wind directions (Fig. 4c), with occasional changes to northerly winds, suggesting that
local circulations had stronger impacts on zonal winds.
Broad peaks in $O_3$ often lasted until late evening hours, while nighttime $O_3$ frequently revealed increases under westerly
winds and could reach daytime concentration levels, which can only be attributed to transport processes. High nighttime $O_3$
was not always accompanied by simultaneous PAN increases, while vice versa, elevated nighttime PAN was also not always
synchronized with those of $O_3$, indicating that they might have originated from distinct sources and processes. $O_3$ levels were
continuously high throughout the spring period, especially during 6-13 May. Despite overall lower levels in the summer
period, two $O_3$ episodes occurred during 7-8 July and 24-25 July, respectively, exhibiting the highest concentrations
observed during the entire campaign (Fig. 2a, Fig. 4e). Compared to $O_3$, PAN displayed much larger day to day variability,
with an evident high PAN episode occurring from 13 to 16 May under southeasterly winds from aloft (Fig. 4f). Summertime
PAN was distinctly lower than that during spring season, with no increases detected during the two high $O_3$ episodes.
Overall, while $O_3$ and PAN revealed highly similar average diurnal variation patterns, their temporal variations often differed
from each other, suggesting that they were determined by distinct transport or formation processes, which will be further
investigated in the following sections.

## 3.2 Impact of local circulation

In previous studies, diurnal variations in $O_3$ and PAN were mainly attributed to local circulations, particularly the
development of the PBL. At pristine mountain sites such as WLG, surface $O_3$ was influenced by free tropospheric air during
nighttime and by boundary layer airmasses during daytime, which resulted in a diurnal cycle with lower daytime and higher
nighttime $O_3$ with very small diurnal variation amplitudes. Despite its high altitudes, NMC is located at the foot of the
northern Nyainqêntanglha Mountains, and thus experienced local circulation distinct from those at WLG. Free tropospheric



air was suggested to be richer in O$_3$ and PAN concentrations and was mixed down upon the rapid development of the
convective boundary layer (CBL) after sunrise, while O$_3$ and PAN concentrations decreased upon the establishment of the
nocturnal boundary layer (NBL), due to the dominance of local boundary layer airmasses during nighttime, which were low
in O$_3$ and PAN, since barely any surface O$_3$ and PAN precursor emission sources existed at NMC, added by effects of dry
deposition (Xu et al., 2018b). The broad O$_3$ peaks that often lasted until late evenings and the frequent events of elevated
nighttime O$_3$ (Fig. 4e) both supported the idea that under favorable meteorological conditions, high surface O$_3$ levels after
sunlight hours could be sustained by continuous downmixing of free tropospheric air. The fact that nighttime O$_3$ could reach
the same level as noontime O$_3$ is why previous studies suggested that physical transport was determining O$_3$ variations at
NMC, while photochemistry played a minor role.
Diurnal variations of O$_3$ and PAN followed their averaged diel pattern on 72% and 75% out of the days with valid records,
respectively. While O$_3$ diurnal cycles revealed more days with such daytime increases during summer (69% in spring vs. 74%
in summer), PAN conformed better to its averaged diurnal cycle in spring (90% in spring vs. 63% in summer), suggesting
that despite being under the same meteorological influences and despite highly similar average diurnal concentration profiles,
O$_3$ and PAN often revealed different variations. O$_3$ and PAN increasing rates (on days with daytime increases) between 7:30
and 10:30 LT both displayed linear correlations to temperature increasing rates (Fig. S4), confirming again that their
morning increases were closely connected to boundary layer development upon radiative heating. Prenoon (6:00 to 12:00 LT)
O$_3$ concentrations also increased with PBLH during both spring (Fig. 5a) and summer (Fig. 5b), however, revealing slightly
distinct slopes during distinct seasons. Weaker prenoon winds that mainly occurred during early morning under low PBLH
conditions were associated with evidently lower O$_3$ concentrations. During early morning hours when PBLH was still low,
strong winds that mostly came from the W-NW direction were associated with O$_3$ concentrations as high as those observed
during noontime in the spring period (Fig. 5a). During the afternoon (12:00 to 18:00 LT), when the CBL has fully
established, O$_3$ hardly displayed any more variation with PBLH (Fig. S5a), indicating that once boundary layer and free
tropospheric air was fully mixed, O$_3$ did not further increase with PBLH. In the summer period, W-NW winds were less
frequent and O$_3$ associated with these winds only increased weakly with PBLH, whereas N-NE winds resulted in more
significant rise in O$_3$ over prenoon hours. Summertime afternoon PBLH was significantly lower than during spring due to
frequent cloudy and rainy conditions, mostly falling into the range of 0.5-1.5 km (Fig. S5b). O$_3$ still increased with PBLH,
however, revealing large variability under the same PBLH, indicating that PBLH was not the deterministic factor for
afternoon O$_3$ levels. PAN did not replicate the variation of O$_3$ with PBLH during prenoon hours, displaying large variability
at lower PBLH and moderate concentration levels under high PBLH. This suggests that free tropospheric O$_3$ levels were
consistently and significantly higher than boundary layer O$_3$ levels, indicating for weak surface formation of O$_3$ (further
discussed in Sect. 3.3), which resulted in significant increases in observed surface O$_3$ upon down mixing. Whereas free
tropospheric PAN or the surface formation of PAN might have had higher variability, which resulted in largely different
responses of PAN with the down mixing of free tropospheric air during afternoon hours.



To investigate what has caused the discrepancies in free tropospheric $O_3$ and PAN over NMC, the variations of surface $O_3$
and PAN with free tropospheric (500-550 hPa) winds during spring and summer are depicted in Fig. 6. At lower wind speeds,
both $O_3$ and PAN typically revealed lower concentrations. With increasing wind speeds, high concentrations of $O_3$ and PAN
were associated with distinct wind directions, in both spring and summer. During springtime, high concentrations of $O_3$ and
PAN both occurred with W winds, however, low $O_3$ and high PAN concentrations were detected under strong S winds.
During summertime, high $O_3$ dominantly occurred with N-NE winds, while PAN mainly revealed elevated concentrations
under S-SW winds. The distinct variation of $O_3$ and PAN with wind speed and wind direction suggests that the
concentrations of both gases might have been impacted by different long-range transport processes, which will be
investigated in the next section.

**3.3 Impact of inter-regional transport and stratospheric-tropospheric exchange**
Potential influence of pollution transport from India and other south Asian countries have been previously reported, which
had potential impacts on the transport of PAN (Xu et al., 2018b). However, the source regions of $O_3$ and PAN in the TP have
not been systematically investigated before. $NO_2$ and CO columns from TROPOMI revealed high concentrations in South
Asian regions south of the TP contrasting to the pristine environment within the TP (Fig. 7). CO was more severe and
widespread outside of the TP during the spring period, while $NO_2$ pollution was more severe during the summer period both
in South Asia and to the east of the TP in China. Inside the TP, $NO_2$ and CO columns were both higher during the summer
period, suggesting that summertime atmospheric circulations might have been more favorable for pollution transport into the
TP. The high-altitude Himalaya mountains along the southern border of the TP is highly effective in blocking out direct
intrusion of South Asian pollution, leading mostly to pollutant accumulation on its southern slope. High resolution satellite
observations clearly reveal high $NO_2$ and CO along mountain and river valleys, indicating that pollution might have
transported into the TP through these passageways. Belts of elevated CO extend from the western side (Kashmir) to the
southeastern corner of the TP, indicating that pollution from South Asia could not directly cross over the Himalayas,
especially not over those regions with very high altitudes, but had entered the TP by crossing either to its west or southeast.
To further identify possible source regions for high $O_3$ and PAN at NMC station, the PSCF for both gases were calculated
for spring and summer, respectively (Fig. 8). Spring time high $O_3$ concentrations were mainly associated with westerly
trajectories, which crossed over North India and Nepal before arriving at NMC (Fig. 8a). Although trajectories associated
with high springtime $O_3$ crossed over vast areas outside the southern TP border, they mainly entered the TP from two
passageways, one from the west and another from the southeast (near the border of Bhutan). Before entering the TP, the
majority of the airmasses associated with high $O_3$ came from higher altitudes (> 6 km), diving downwards to heights of 3-
6 km or even < 3 km near the southern border of the TP, and then entering the TP mainly from the west or south (Figs. 9a1-
3).  Aside from that, trajectories from the NW mostly travelling within 0-6 km (above ground level) were also associated
with high springtime $O_3$. High springtime PAN, however, was only associated with trajectories crossing over South Asia and



entering the TP from the southeastern border. In addition, airmasses from the Indian Ocean that travelled within 0-3 km and
crossed over Bangladesh and Bhutan were also associated with high PAN, while not with high $O_3$ (Fig. 8b and Figs. 9b1-3).
During summer, the PSCF of $O_3$ revealed a largely different distribution from that in spring. High altitude westerly airmasses
that entered the TP from the west in spring have not been seen in summer, while airmasses sweeping along the southern
border of the TP (Nepal and northern India) at altitudes below 6 km and approaching NMC from its south were still partly
associated with high $O_3$ during summer (Fig. 8c and Figs. 9c1-3). Southerly low altitude (0-3 km) maritime airmasses that
travelled over Bangladesh and Bhutan before entering the TP were also sometimes linked to high $O_3$ at NMC. However, the
major summertime $O_3$ source regions were located to the north of NMC, including southern Xinjiang province, Northern
Tibet and western Qinghai Province (Fig. 8c). High $O_3$ was mostly associated with low altitude airmasses from the NW and
N directions (Fig. 9c1). Summertime PAN was only rarely associated with northerly airmasses, but mostly linked to westerly
trajectories that travel along the southern TP border (mostly within 0-3 km, small parts within 3-6 km altitude, Figs. 9d1-2)
and southerly trajectories travelling over Bangladesh and Bhutan within 3 km altitude (Fig. 9d1).
Thus, $O_3$ and PAN revealed distinct source regions in both spring and summer, while they also shared some common source
regions. This explains why despite highly similar diel variation patterns, the day-to-day variation was often different between
the two oxidants. Overall, springtime synoptic conditions resulted in a relatively monotone origin of airmasses at NMC,
mostly favoring the subsidence of high altitude airmasses under westerly airflows, which were rich in both $O_3$ and PAN.
With the onset of the South and East Asian Monsoon during summer, circulations drastically changed and resulted in
influences of various distinct airmass origins at NMC. These vastly different airmass origins also exhibited completely
different $O_3$ levels, with those originated in the north exhibiting even higher $O_3$ levels than those observed during springtime
and southerly airmasses revealing much lower $O_3$ levels than during springtime. PAN, however, was more linked to westerly
and southerly airmasses during summer.
Aside from changes in air mass origins at NMC, seasonal variations in large scale synoptic conditions were also
deterministic of STE and the overall spatial distribution of $O_3$. Since the ERA5 reanalysis data has only considered
simplified stratospheric $O_3$ chemistry and the physical transport of $O_3$, the $O_3$ mixing ratio in the ERA5 dataset is a good
indicator for the investigation of stratospheric influences. During the spring period, the averaged ERA5 500 hPa $O_3$ revealed
relatively lower mixing ratios in the TP region (especially in southeast TP) and higher mixing ratios outside the TP in the
latitude band between 15 and 25°N. As was shown in previous studies, the downward transport of stratospheric $O_3$ and its
distribution is closely linked to the location of the subtropical jet stream (Xu et al., 2018a) , which is typically located above
the TP during the spring period (Fig. 10c1).  Due to large scale circulations, lower stratospheric $O_3$ is typically high in polar
regions, decreasing with latitude and reaching its lowest level in the equatorial belt (Fig. 10c1). Deep stratospheric intrusion
and $O_3$ subsidence often occur along the lower edge of the subtropical jet stream, which is a slope extending from the lower
stratosphere (150 hPa) between 38 to 42°N down to the middle or upper troposphere below 28°N. STE processes are
especially promoted by fronts, which are accompanied by large scale subsidence of cold air from above (Stohl et al., 2003).
STE mostly increased the $O_3$ levels in Southeast Asia to the west and south of the TP, which in turn could enhance $O_3$ at



NMC through the westerly airmass transport passage (Figs. S6). Direct STE influence was also frequently observed during
the spring period (on 5-8, 13, 23, 31 May and 3, 5 and 9 Jun, Figs. S7-13), with NMC frequently located near low pressure
troughs behind cold fronts. These STE events were typically associated with high $O_3$ and low PAN concentrations, except
for the 13 May, when stratospheric $O_3$ was transported to lower latitudes and then back to NMC via southwesterly winds,
which also carried along high PAN concentrations, suggesting that NMC experienced aged stratospheric airmasses. During
the summer period, with the northward shift of the subtropical jet stream, the high lower stratospheric $O_3$ concentrations
were also confined within higher latitudes. 500 hPa ERA5 $O_3$ revealed a clearly distinct distribution from that during spring,
displaying higher $O_3$ levels north of NMC (>30 °N) and much lower ones in the tropical region. Thus, under the prevailing
southerly winds during the summer season, airmasses with lower stratospheric $O_3$ contents are transported to NMC. However,
during two episodes on 7-9 and 21-25 Jul, northerly cold airmasses in front of 500 hPa high pressure systems over the TP
brought stratospheric $O_3$ down to the norther TP regions and transported them within lower altitudes to NMC, resulting in
surface $O_3$ levels even higher than those during springtime (Figs. S14-15), while PAN did not reveal significant increases.
Statistically, $O_{3,ERA5}$ only explained 0.1% of the observed daytime $O_3$ day-to-day variability during spring (r=0.033),
however, explained 22% of the summertime $O_{3,NMC}$ variability (r=0.47), contributing on average 10% during the entire
observation period (Fig. S16), which was overall in accordance with previous results reported in Yin et al. (2017). It is also
worth noting that observed $O_3$ at NMC was typically higher than the 550 hPa ERA5 $O_3$ mixing ratio, especially during spring
and early summer (Fig. 10d). During the entire observation, stratospheric $O_3$ transport explained 83% of the observed
daytime $O_3$ concentration ($O_{3,ERA5}/O_{3,NMC}$), with a lower contribution during spring (77%) and a higher one during summer
(88%). This suggests that despite the small contributions of STE to the day-to-day variability of observed $O_3$, the overall
daytime $O_3$ concentration was mainly maintained by the long-range transport of stratospheric $O_3$ (as opposed to direct strike
of stratospheric $O_3$ during deep STE intrusions into the PBL). Additionally, the unexplained $O_3$ concentration might indicate
for photochemical $O_3$ formation aside from pure physical transport. However, whether it was caused by local photochemical
production or the long-range transport of photochemically produced $O_3$ still requires further investigation.

**3.4 Impact of local photochemistry**
As was already manifested, $O_3$ has its natural sources and is more affected by STE processes at high altitude locations such
as NMC. $O_3$ is highly reactive and can be easily depleted in regions with high $NO_x$ and VOCs emissions, however has a
relatively longer lifetime in pristine background areas and can by directly or indirectly transported (transport of its precursors)
over large distances, affecting $O_3$ levels at remote locations (Xu et al., 2018a). The impact of local photochemistry on the
budget of $O_3$, however, was often under debate in previous studies conducted in background areas of the TP. Under such
pristine atmospheric conditions, it was manifested that $O_3$ production was strongly $NO_x$-limited, with $NO_x$ concentrations
being the key factor determining whether $O_3$ was net produced or destructed in local photochemistry (Ma et al., 2002).
However, the lower detection limit and precision of commercial instruments can hardly meet the needs for $NO_x$
measurements in such clean environments, which made it difficult to determine whether there has been net $O_3$ formation. At





higher altitudes, PAN has a long lifetime and can be transported over long distances. PAN measurements have been
previously conducted at Mt. Waliguan (Northeastern TP) in 2006 (Xue et al., 2011) and at NMC station in the springs and
summers of 2011 and 2012 (Xu et al., 2018b). At both sites, PAN contributed substantially to reactive nitrogen and acted as
a good indicator for regional and long-range transport of polluted air plumes. The photochemical formation of PAN requires
the presence of peroxyl acetyl radical and $NO_2$. The former is only formed in photochemical reactions of its precursor
OVOCs, which are predominantly emitted within the boundary layer, while the latter is also mostly emitted near surface,
with the exception of lightning processes. Altogether, the formation of PAN in comparison with $O_3$ is more favored near
surface and has no natural sources. Nevertheless, the impact of local PAN formation versus those of transport to observed
concentration levels was not discussed before due to the lack of its precursor measurements.
To evaluate the contribution of local photochemistry to observed $O_3$ and PAN, simulations were performed using an MCM-
based box model for the period of 1 to 21 May, when VOCs measurements were available. Observed $O_3$ revealed much
larger fluctuations than those obtained from all three simulation scenarios, which respectively used measurement constraints
on OVOCs, $NO_2$ or both of them (Fig. 11a). With constraints on $NO_2$, modelling results revealed significant daytime
increases, indicating positive local net photochemical formation of $O_3$ (Fig. 12a). However, when $NO_2$ was unconstrained,
modelled $O_3$ concentrations were significantly lower and displayed very small variability, with very small positive net $O_3$
production during the morning and mostly negative ones during the day (Fig. 12a). Nevertheless, none of the simulations
could reproduce the large variability and steep morning increases within observed $O_3$, with OVOCs and $NO_2$ both
constrained by measurements, modelling results could only explain 28±19% (5-66%) of the observed daytime increases (Fig.
11b), while even less could be explained when only OVOCs or $NO_2$ was constrained (3±6% and 21±14%, respectively).
Days with relatively stronger local photochemical $O_3$ formation were not necessary days with high observed $O_3$, while in
return days with high $O_3$ were also often associated with weak photochemical net $O_3$ formation. This indicates that physical
transport and mixing processes were determinative of $O_3$ diel cycle as well as the day-to-day $O_3$ variability, while local
photochemistry further added to the daytime $O_3$ burden. Additionally, intercomparison among simulations also confirmed
the high sensitivity of $O_3$ formation towards $NO_x$ and the relatively weaker sensitivity to VOCs in such a pristine
environment.
Simulated PAN levels under $NO_2$ constraints were, however, significantly higher than observed PAN concentrations,
especially when OVOCs and $NO_2$ were both constrained. However, when $NO_2$ was unconstrained, PAN concentrations were
mostly underestimated by simulations. Thermal decomposition of PAN was very weak under low springtime temperatures
and net photochemical PAN formation rates were positive under all simulation scenarios, however, only $NO_2$ constrained
cases revealed strong formation throughout daytime hours (08:00-20:00 LT) while unconstrained $NO_2$ simulations only
displayed a very weak morning time (07:00-9:00 LT) photochemical formation. Only $NO_2$-constrained simulations
overestimated PAN concentrations by a factor of 1.8 on average, however, could reproduce observed daytime PAN
increments by 94±84%. Additionally, days with high simulated PAN photochemical production (4-6 and 13-17 May)
corresponded to episodes with elevated observed PAN concentrations, which indicates that photochemical formation of PAN





were determinative of its day-to-day variability. Compared to $O_3$, PAN was sensitive to concentrations of both OVOCs and $NO_2$, since some of the OVOCs are direct precursors of PA radicals, which combine with $NO_2$ in PAN formation. Still, $NO_2$ was more decisive of the overall $O_3$ and PAN production, since without its constraint, $O_3$ net loss and negligible PAN net formation would be yielded.

It should also be noted that both observed $O_3$ and PAN were not necessarily formed within the local boundary layer, since springtime winds in the TP are very strong, especially during daytime. Due to its relatively long lifetime, PAN might have been formed on the transport pathway to NMC, while $O_3$ might undergo both destruction and production during airmass transport. This might partly explain why PAN formation was overestimated by simulations representing surface conditions. But overall, it could be concluded that $O_3$ was mainly determined by physical transport, particularly STE processes, while PAN was largely determined by local photochemistry and that along the transport passageway. Fresh STE plumes reaching NMC from the north where PAN concentrations result in depleted surface PAN, while relatively aged STE airmasses crossing over polluted regions of Indo-Gigantic Plain led to simultaneous enrichment in surface $O_3$ and PAN. The high sensitivity of $O_3$ and PAN formation towards $NO_x$ indicates that increased natural emission of $NO_x$ under global warming, enhanced anthropogenic emissions of $NO_x$ within the TP region due to the development of highways and transportation as well as increased transport input from South Asia might greatly enhance $O_3$ and PAN formation in background regions, while increased VOCs emissions and regional transport promotes PAN formation more than that of $O_3$.

## 4 Conclusions and implications

In this study, continuous measurements of $O_3$ and PAN as well as its precursors were conducted during spring and summer at a highly pristine high-altitude site in the southern TP (NMC station) to investigate the factors determining their variations. Due to the local topography, surface observations at NMC reflects free tropospheric air conditions during daytime and nocturnal boundary layer conditions during nighttime. Both $O_3$ and PAN revealed steep increases after sunrise and reached a flat plateau during daytime. While averaged diurnal variations of $O_3$ and PAN highly resembled each other, their day-to-day variations were often different, suggesting that they might have been influenced by distinct physiochemical processes.

Backward trajectory modelling and PSCF analysis revealed distinct source regions connected to high $O_3$ and PAN. During spring, airmasses rich in $O_3$ were mainly associated with high altitude westerly airmasses that either entered the TP from the west of from the south, while PAN was only rich in westerly airmasses that transported along the polluted regions in North India and Nepal before entering the TP from the south or in southerly maritime airmasses that crossed over polluted South Asian regions before entering the TP. During the summer monsoon season, airmasses from the north were associated with the highest $O_3$ levels, while westerly and southerly airmasses revealed lower $O_3$ levels. Elevated PAN concentrations, however, were still linked to westerly and southerly airmasses crossing over polluted South Asian regions. $O_3$ at NMC was strongly influenced by STE, which brought down high stratospheric $O_3$ concentrations from the southwest route during spring and from the northwest during summer, explaining 77% and 88% of the observed $O_3$ level in spring and summer,



respectively. PAN concentrations were, however, typically lower in airmasses with strong stratospheric influence, except if
they transported over polluted regions south of the TP.
Photochemistry resulted in positive net formation of both $O_3$ and PAN. While only 28±19% of the observed daytime growth
in $O_3$ could be explained by photochemical simulations, the daytime growth of PAN was highly overestimated by the model
if OVOCs and $NO_2$ were both constrained. Photochemistry was not the factor determining the day-to-day variability of $O_3$,
however, explained observed PAN variabilities well. While both $O_3$ and PAN formation were highly sensitive to $NO_x$ levels,
PAN was also quite sensitive to VOCs concentrations. Therefore, future concentrations of $O_3$ and PAN over the TP may be
significantly impacted by increases in the concentrations of $NO_x$, VOCs, and other precursors, which either originate from
the surrounding regions (in particular South Asia) or from anthropogenic and natural sources within the TP. Special attention
should be addressed to PAN, which is mostly determined by photochemical processes sensitive to both $NO_x$ and VOCs and
can be transported over very long distances.

**Data availability**. The data used in this study are available on the @Tibet ftp server (http://at-tibet.quickconnect.cn/) and
can be applied for upon request to the corresponding authors (zhanggen@cma.gov.cn and c.ye@pku.edu.cn)

**Author contributions.** WX and CY designed the experiment and led the research. WX, GZ, CY, YW, YZ, YB, WL, XX
were responsible for the maintenance of trace gas and meteorology measurements in the experiment and WX, YZ and YW
processed the data. WX analyzed the data and wrote the paper with help from XZ, XX and GZ.

**Competing interests**. The authors declare that they have no conflict of interest.

**Acknowledgments, Samples, and Data**
This work is supported by the National Natural Science Foundation of China (41875159, 42175127, 42275127,
42075112, and 42105110) and the Natural Science Foundation of Beijing (8222078).

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



**Table 1** Statistics of trace gases (including $O_3$, PAN, $NO_2$, CO, OVOCs, aromatics, alkanes and alkenes), photolysis rates ($jO^1D$ and $jNO_2$), meteorological variables, as well as the Y index

| Variable | Unit | Spring | | | Summer | | |
|---|---|---|---|---|---|---|---|
| | | all | day (8:00-20:00) | night (20:00-8:00) | all | day | night |
| $O_3$ | ppb | 59.8±13.4 | 67.8±9.0 | 52.2±12.4 | 53.6±13.2 | 58.3±12.5 | 48.8±12.1 |
| PAN | ppb | 0.27±0.08 | 0.30±0.07 | 0.24±0.07 | 0.20±0.05 | 0.21±0.05 | 0.18±0.05 |
| $NO_2$ | ppb | 0.12±0.05 | 0.11±0.07 | 0.13±0.04 | 0.09±0.05 | 0.08±0.03 | 0.10±0.06 |
| CO | ppb | 108±26 | 108±16 | 107±33 | 117±29 | 116±33 | 118±24 |
| $CH_4$ | ppm | 1.890±0.024 | 1.884±0.012 | 1.895±0.030 | 1.886±0.021 | 1.883±0.017 | 1.887±0.024 |
| OVOCs | | 2.49±2.16 | 3.10±2.60 | 1.88±1.34 | | - | - |
| Aromatics | Propy-Equiv. ppbC | 0.56±1.29 | 0.61±1.74 | 0.51±0.55 | - | - | - |
| Alkanes | | 0.47±0.50 | 0.48±0.59 | 0.46±0.40 | - | - | - |
| Alkenes | | 0.59±0.57 | 0.71±0.68 | 0.47±0.39 | - | - | - |
| $jO^1D$ | $10^{-7}$ s$^{-1}$ | - | 277±183 | - | - | 275 | - |
| $jNO_2$ | $10^{-4}$ s$^{-1}$ | - | 70±27 | - | - | 66±29 | - |
| Temperature | °C | 4.2±4.1 | 6.6±3.4 | 2.0±3.5 | 9.3±3.5 | 10.9±3.5 | 7.7±2.8 |
| RH | % | 50±19 | 41±18 | 59±17 | 61±19 | 55±19 | 68±17 |
| Cumulated Rain | mm | 1.0 | 0.8 | 0.2 | 37.3 | 24.1 | 13.2 |
| Wind Speed | m s$^{-1}$ | 4.0±2.6 | 4.8±2.3 | 3.3±2.6 | 3.9±2.3 | 4.2±2.2 | 3.6±2.4 |
| Y index | - | 1.7±0.9 | 2.1±0.9 | 1.4±0.6 | 0.8±0.3 | 0.9±0.4 | 0.7±0.3 |



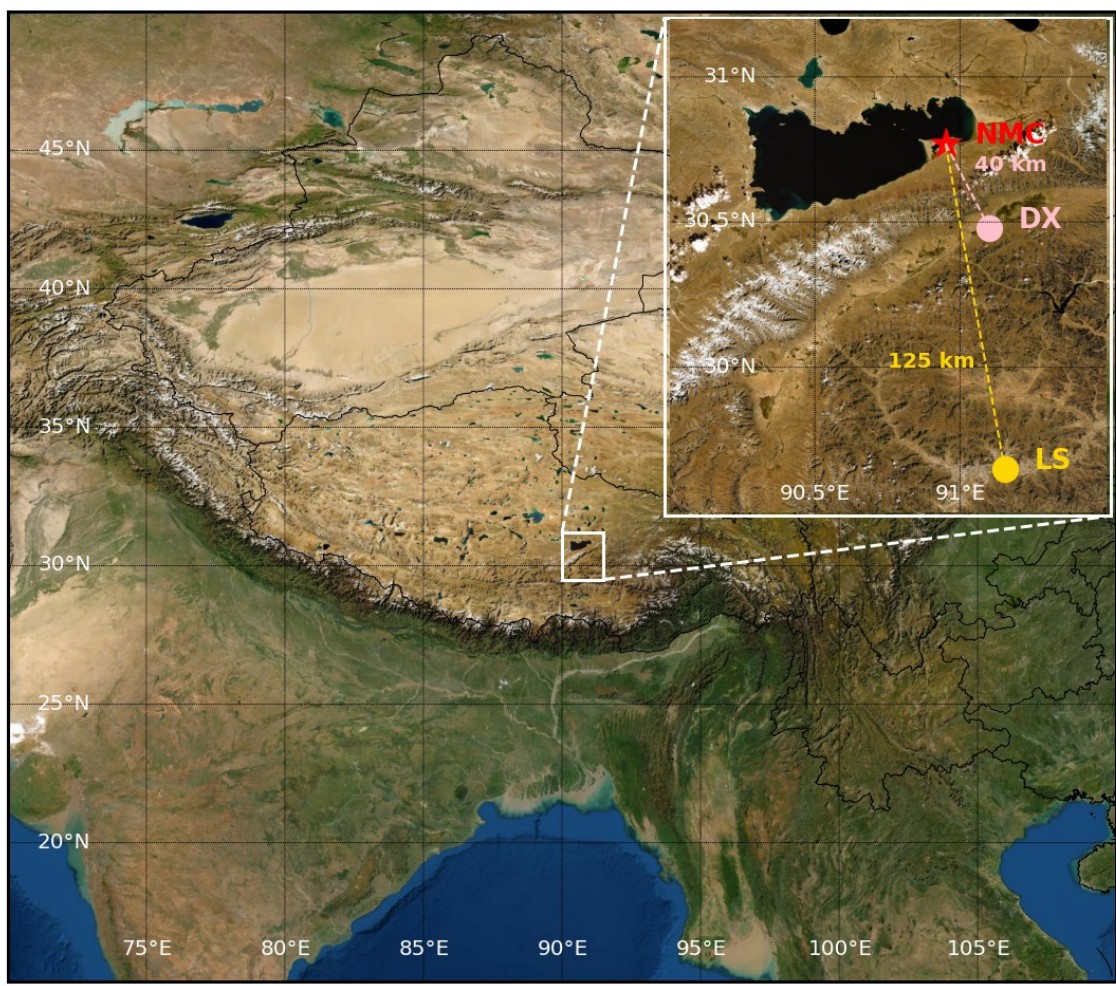

**Figure 1. Map displaying the locations of the Nam Co site (NMC), Dangxiong (DX) county and Lhasa city (LS). This figure was draw based on Map service in ArcGIS World Imagery (https://doc.arcgis.com/en/data-appliance/6.4/maps/world-imagery.htm)**



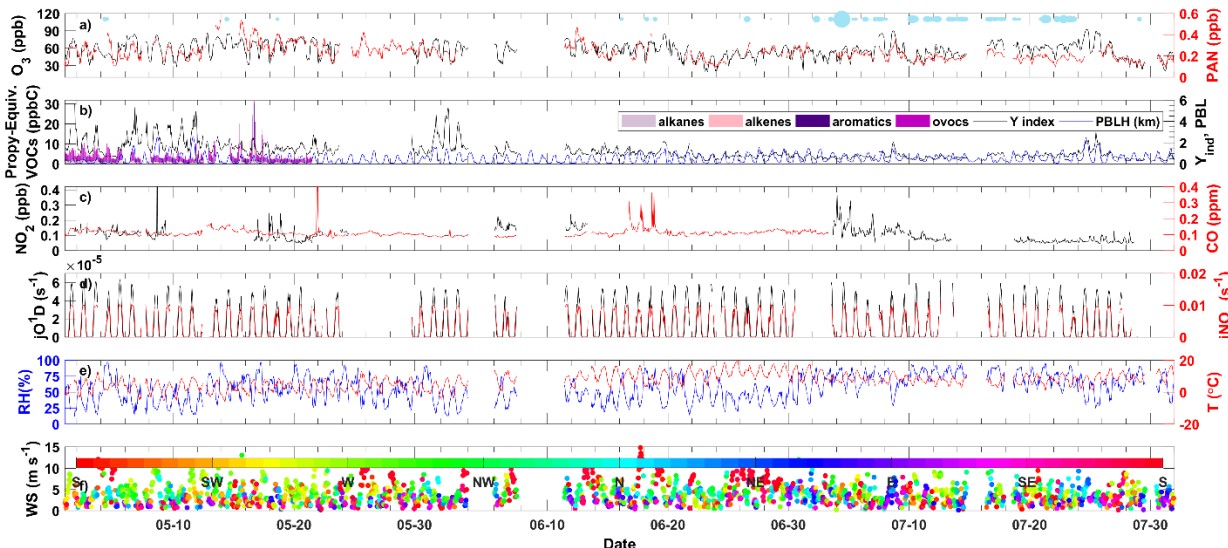

590

**Figure 2 Timeseries of a) O₃, PAN, b) NO₂, CO, c) jO¹D, jNO₂, d) RH, T, e) wind speed and wind direction during the Nam Co campaign from 1 May to 31 Jul.**

593

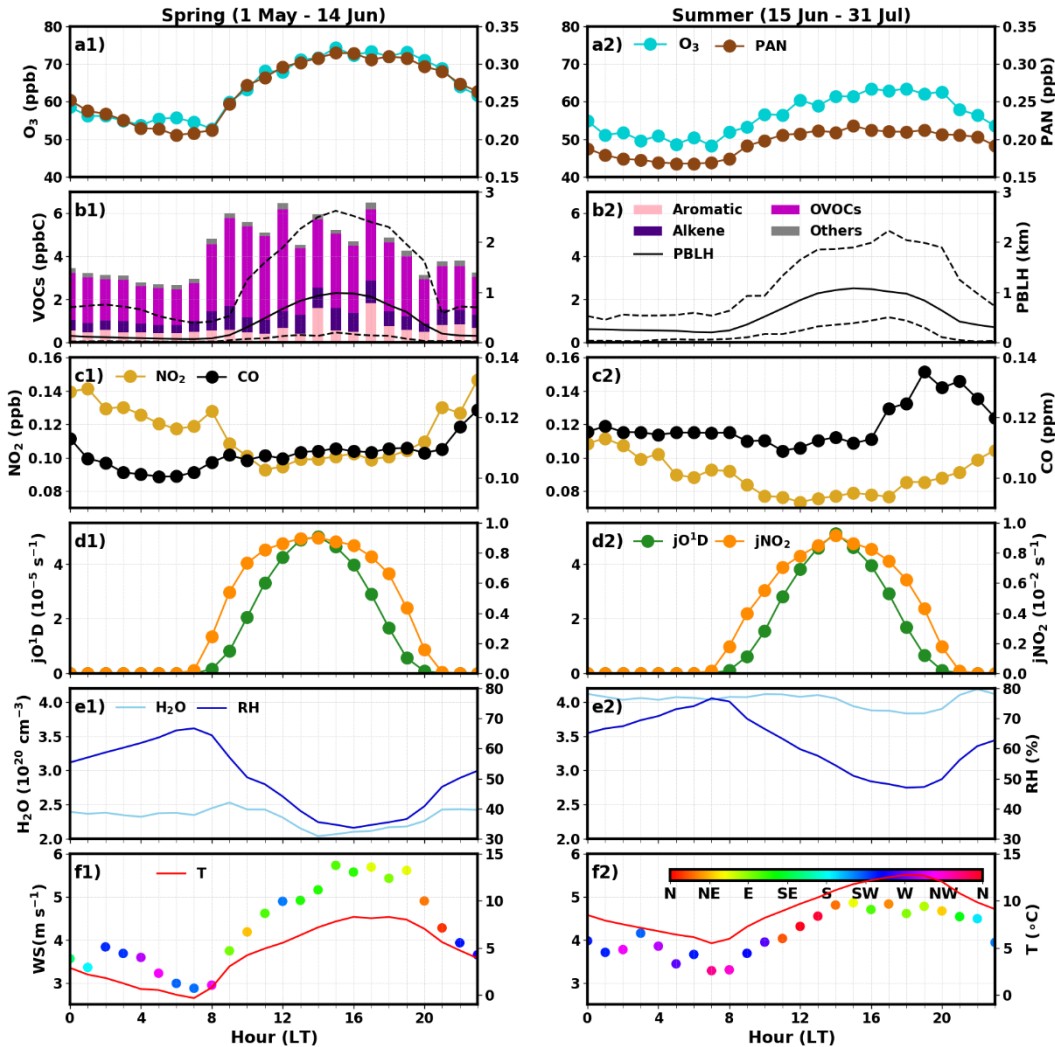

**Figure 3.** Averaged diurnal variations of a) $O_3$, PAN, b) VOCs, PBLH (solid line black line: average value, dashed black lines: minimum and maximum value) c) $NO_2$, CO, d) $jO^1D$, $jNO_2$, e) $H_2O$, RH and f) temperature, wind speed and wind direction during the 1) spring and 2) summer period, respectively.



**Figure 4. Season-diurnal variations of a) ω wind, b) U wind, c) V wind, d) PBLH, e) surface O$_3$ and f) PAN between 1 May and 31 Jul 2019 at Nam Co.**







**Figure 5. Variation of prenoon (6:00-12:00) O₃ (a,b) and PAN (c,d) with PBLH (from ERA5 reanalysis data) during spring (a,c) and summer (b,d) periods, with wind speeds and directions indicated by sizes and colors of scattered dots (precipitation associated data points excluded).**





608

Figure 6. Variation of springtime (a,b) and summertime (c,d) O$_3$ (a,c) and PAN (b,d) concentrations with 2m wind speeds and 500-550 hPa wind directions from ECMWF ERA5 data. Gray shading represents the relative occurrence frequency of wind directions.

611



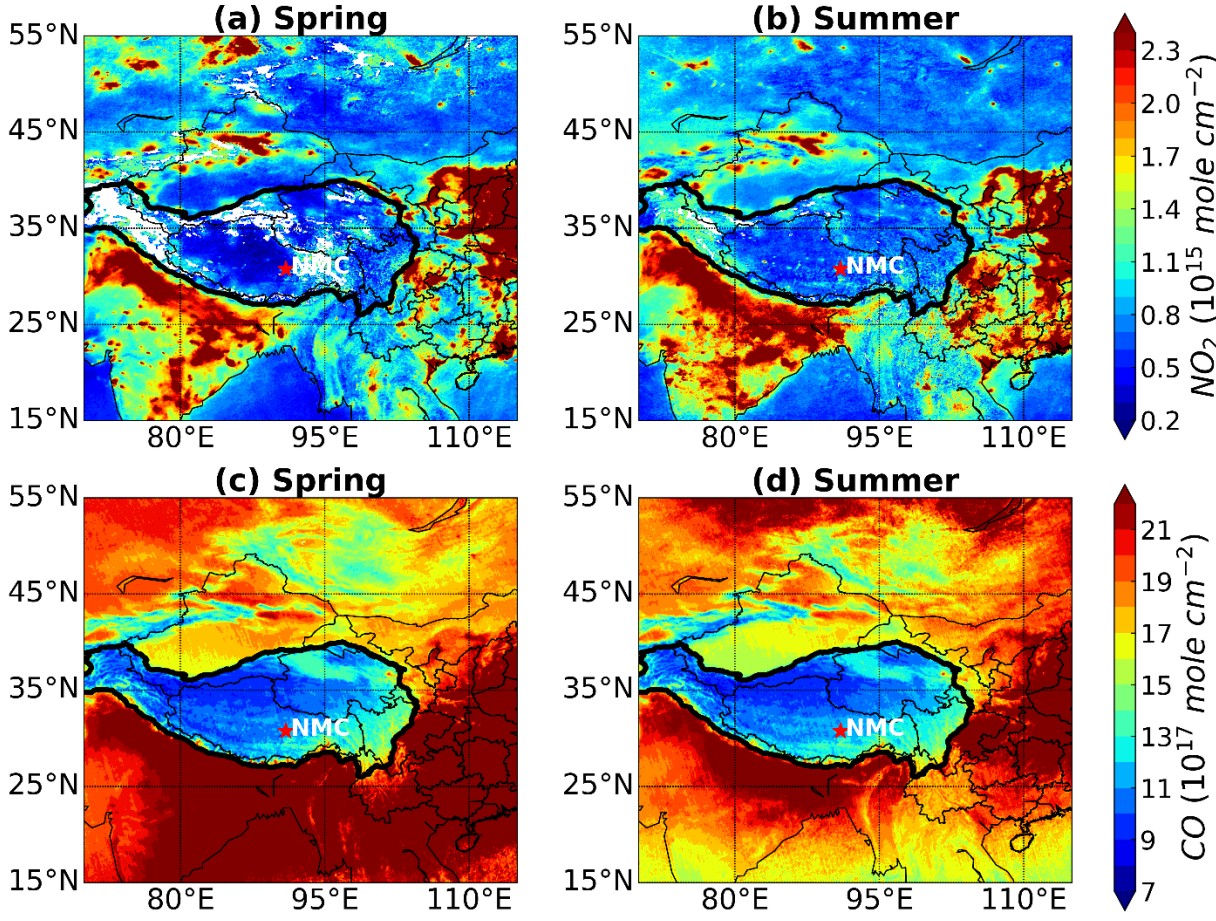

**Figure 7.** **TROPOMI NO$_2$ (a,b) and CO (c,d) column concentration distributions averaged over spring (a,c) and summer (b,d)**
**periods.**



**Figure 8. Potential Source Contribution Function (PSCF) of O₃ (a,c) and PAN (b,d) during spring (a,b) and summer (c,d) periods.**



**Figure 9. Residence time of trajectories associated with O₃ (a,c) and PAN (b,d) above their respective 67ᵗʰ percentiles during spring (a,b) and summer (c,d) periods within height ranges (above ground level) of 1) 0-3 km, 2) 3-6 km and 3) >6 km.**





Figure 10. Distribution of a) ERA5 500 hPa $O_3$ mixing ratio, geopotential height (white contour lines) and winds (black arrows), b) 250 hPa potential vorticity, geopotential height and winds, c) cross-section of $O_3$ mixing ratio, u winds (white contour lines), v winds and vertical velocity (black arrows) at the longitude of Nam Co station and d) the comparison between daytime ERA5 550 hPa and observed $O_3$ mixing ratio at Nam Co.







**Figure 11. a)** Observed (black) and modelled O₃ using constraints on OVOCs (green), NO₂ (blue) and both (red), **b)** percentage of observed daytime O₃ concentration increment ($\Delta O_{3,obs}$) that can be explained by those modelled under different constraints ($\Delta O_{3,mod}$), **c)** observed (black) and modelled PAN under OVOCs (green), NO₂ (blue) and both constraints (red), **d)** percentage of observed daytime PAN concentration increment ($\Delta PAN_{obs}$) that can be explained by those modelled under different constraints ($\Delta PAN_{mod}$).



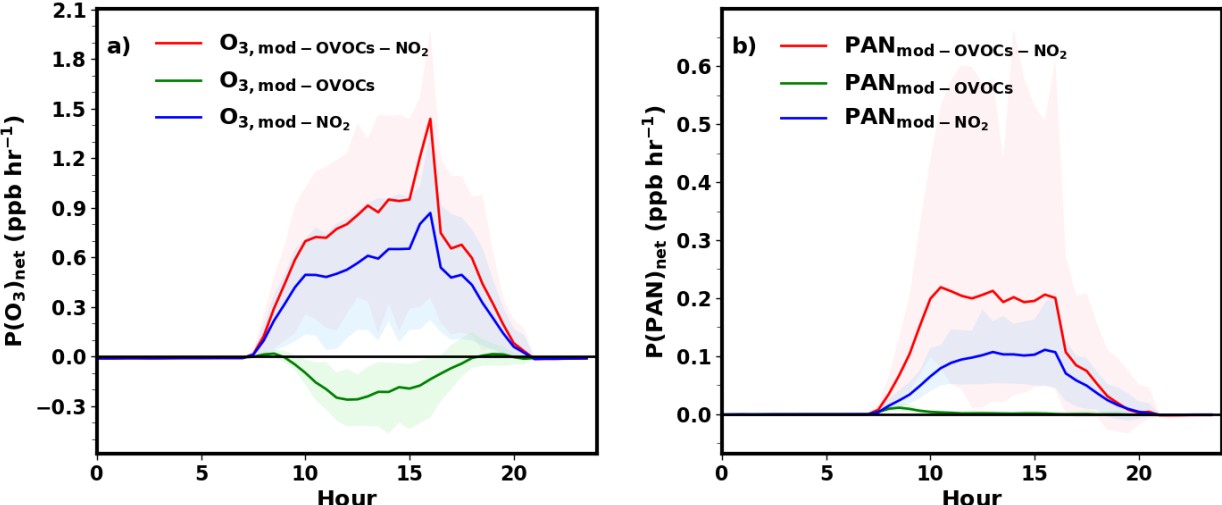

**Figure 12. Net production rate of a) $O_3$ and b) PAN simulated under OVOCs (green), $NO_2$ (blue) and OVOCs+$NO_2$ combined measurement constraints. Shaded areas represent calculated ranges of 5$^{th}$ to 95$^{th}$ percentiles.**