# Peer review of "O3 and PAN in southern Tibetan Plateau determined by distinct physical and chemical processes"

_EGUsphere, 2023_

## Author Comment (AC1)

**Response to Reviewer #1:**

**Reviewer #1:** The authors conducted a study to investigate the variations in tropospheric ozone ($O_3$) and peroxyacetyl nitrate (PAN) and their precursors at a high-altitude site in the southern Tibetan Plateau. They found that variations in $O_3$ and PAN were controlled by distinct processes, with $O_3$ being mostly determined by stratosphere-troposphere exchange (STE), while PAN being mostly influenced by tropospheric chemistry and transport. While local surface $O_3$ and PAN both revealed net photochemical production during daytime, the variability of PAN was more controlled by photochemical formation processes than $O_3$. Overall, the manuscript is well organized and fluent in language. The analysis methods and results are scientifically sound. To my knowledge, this is the first study in Tibet presenting simultaneous high time resolution measurements of $O_3$, PAN as well as its precursors, which are highly valuable data worth publishing. Additionally, this is the first evaluation of PAN formation in Tibet, which also fills in a gap of knowledge. Some minor issues need to be addressed before the manuscript can be accepted for publication.

**Response:** We thank the reviewer for carefully going through our manuscript and for acknowledging the value of this research. In the following we will provide point-to-point responses to the detailed comments and suggestions.

**Reviewer #1:** Minor comments:

1. Lines 136-156: A slight concern on the PSCF analysis is that this is typically more suitable for the source analysis of atmospheric components with longer lifetimes, while PAN can have a long-lifetime in the free troposphere, $O_3$ seems to be more short-lived.

   **Response:** We thank the reviewer for raising this concern. $O_3$ indeed has relatively short lifetime in polluted regions, mainly due to its consumption by anthropogenic VOCs and $NO_x$ emissions. However, at remote and pristine high altitude mountain sites such as those in Tibet (including the Lake Nam Co, Mount Waliguan and Xianggelila sites), $NO_x$ levels are extremely low (~50 ppt on average) and concentrations of VOCs are also far lower than in polluted urban regions, which allows $O_3$ to have a relatively longer lifetime. The global lifetime of $O_3$ was estimated to be ~22 days (Goldberg et al., 2015), however, Bates and Jacob (2020) introduced an expanded odd oxygen family in the calculation of $O_3$ lifetimes and reached the conclusion that the global mean $O_3$ lifetime can reach 73 days. Previous studies at Mt. Waliguan have shown that $O_3$ can be transported over long distances from Southeast Asia to the northeastern edge of the Tibetan Plateau (Xu et al., 2018). The PSCF analysis was also used together with surface $O_3$ observations, which identified free tropospheric contributions in the Northwest sector and high anthropogenic contributions

from the Southeastern sector. Thus, using the PSCF analysis on surface $O_3$ in Nam Co was believed to be adequate.

2. Lines 198-205: The authors only provided Propy-Equivalent VOCs concentrations, however, to put observations into context with those in other studies, it might also be necessary to provide direct VOCs concentrations in ppb or ppbC. How do VOCs levels compare with previous observations in Tibet or similar high-altitude sites around the world? Since the authors discuss $O_3$ and PAN photochemistry afterwards, this might be important for understanding the difference or similarities in their results with those in literature.

   **Response:** We thank the reviewer for this suggestion. We added VOCs concentrations (in ppbv) and included a brief comparison between VOCs levels observed in this study and those obtained at other mountain sites, and added a short discussion on sources of VOCs at Nam Co:

   "Alkanes, alkenes and aromatics observed at the Dinghu mountain (1000 m a.s.l.) background site in southern China were 48, 40 and 29 times of those observed at the Nam Co site (Wu et al., 2016). Those in the Rocky Mountain National Park (3498 m a.s.l.) were 1.2, 3.6 and 1.3 times of those observed in this study (Benedict et al., 2019), while those observed during summertime (1994-1996) were 0.9, 14 and 1.6 times of that those in Nam Co (Ma et al., 2002), revealing the extremely low primary VOCs emissions at our site. However, OVOCs concentrations at Nam Co were 1.3 times of those observed in the Rocky Mountains, while only 0.24 times of those previously observed at Mt. Waliguan (Mu et al., 2007), indicating that airmasses in the TP were strongly photochemically aged due to the strong radiation and high oxidative capacity at such high altitudes, with additional influences from natural sources such as plant emissions or animal excrement (mostly from yak and sheep). At Nam Co, concentrations of isoprene and its oxidation products (e.g. MVK and MACR) were very low (0.034 ppb in total), with OVOCs being mostly dominated by formaldehyde, acetaldehyde and acetone (3.2 ppb in total), which have shown elevated concentrations over animal excrement (Mu et al., 2007)."

3. Lines 212-215: Again, how do $NO_2$ concentrations compare with previous observations in Tibet (if there are any)? The authors mention that there are natural and anthropogenic emissions in $NO_x$, can you name the detailed sources influencing $NO_x$ variations at Nam CO? How do these sources impact the diurnal variation of $NO_x$?

**Response:** Thank you for the suggestion, we added a brief comparison with previous observations made at Mt. Waliguan in the northeastern Qinghai-Tibetan Plateau. Additionally, we pointed out the possible sources of NOx at Nam Co.

"$NO_x$ levels were only scarcely reported for remote high-altitude locations, mainly due to instrument limitations. $NO_2$ levels were only slightly higher than the average $NO_2$ and NO level reported at Mt. Waliguan during summer 1994-1996 (0.048±0.017 ppb) based on filter-pack sampling and spring 2003 (0.043±0.069 ppb) based on chemiluminescence (Ma et al., 2002; Wang et al., 2006). Since Nam Co is located far away from anthropogenic emission sources, $NO_x$ levels here are mainly determined by natural emissions, such as those from soil microbial activities or lightning processes. Additionally, a latest study has proposed that lakes in the TP are strong $NO_x$ emission sources of $NO_x$ (Kong et al., 2022). "

4. Lines 252-260: Here the authors conclude that $O_3$ and PAN decreases after sunset are caused by dry deposition as well as the cutoff of free-tropospheric input, indicating that local formations $O_3$ and PAN are weak. However, in the later sections, PAN formation at least was revealed to be strong. Is there a contradiction?

**Response:** We thank the reviewer for the good question. Lines 252-260 mainly summarizes the conclusions in a previous study on $O_3$ and PAN at Nam Co. Free tropospheric airmasses were suggested to be richer in PAN, because they were not influenced by dry deposition compared to near surface airmasses. Additionally, it was speculated in their study that PAN precursors observed at Nam Co mainly came from transport and photochemical conversions on their transport pathway, rather than local emissions and photochemical production.

Results in our study, however, have revealed that near surface PAN formation can be strong under observed $NO_x$ and VOCs levels, which even overexplains the observed daytime increases. However, wind speeds were very strong at Nam Co, especially during daytime. Airmasses crossing over Nam Co only had very short residence times. Thus, observed daytime PAN concentrations were not necessarily representative of in situ photochemical formation, but were also greatly influenced by free tropospheric airmasses and PAN formation therein, which partly explained the overestimation of the box-model. From Fig. 12b in the manuscript, it was noted that photochemical PAN formation ends after sunset, thus the overestimated PAN in the box model after sunset mainly resulted from the fact that nighttime PAN loss terms (such as dry deposition) could not consume all the overestimated PAN formed over daytime within the model. Thus, the gradual decrease in observed PAN concentrations after sunset

did not indicate that surface formation of PAN was weak, it only suggests that with no photochemical formation and without additional downmixing from the free troposphere, near surface PAN would be partly lost through dry deposition.

5. Lines 258-264: PAN often revealed high concentrations at moderate PBLH, which the authors explained as *"Whereas PAN in the free troposphere might have had higher variability, which resulted in largely different enhancements of PAN upon down mixing.".* Might it be that local PAN formation was controlling the large variability of PAN?

   **Response:** The reviewer put forward a very good point. The large variability of PAN might have been the combined effect of high variability in free tropospheric PAN loading and high variability in PAN formation within the observed airmass. The fact that only moderate concentrations of PAN were observed at high PBLH suggests that intensive downmixing of free tropospheric air might have had a dilution effect on local PAN. Thus we modified this part as:

   "Whereas free tropospheric PAN or the surface formation of PAN might have had higher variability, which resulted in largely different responses of PAN with the down mixing of free tropospheric air. Intensive downmixing for free tropospheric air under high PBLH might have even diluted boundary layer PAN concentrations."

Technical comments:

1. Line 18 and later in text: "physiochemical" should be replaced with be "physicochemical".

   **Response:** Thank you for pointing that out, we changed "physiochemical" to "physicochemical" throughout the entire manuscript.

2. Line 28: "could by" should be "could be explained by"

   **Response:** Thanks for noticing, we corrected this typing error according to your suggestion.

3. Line 45: Change "nearly proportional to its OVOCs precursors" to "nearly proportionally to its OVOC precursors".

**Response:** Thank you for the suggestion, we made according changes.

4. Line 440: "reflects" should be "reflect"

    **Response:** Thanks, this mistake has been corrected.

5. Line 446: The "of" should be "or" in "from the west of from the south"

    **Response:** Thank you for pointing that out, we corrected it to "or".

References:

Bates, K. H., & Jacob, D. J. (2020). An Expanded Definition of the Odd Oxygen Family for Tropospheric Ozone Budgets: Implications for Ozone Lifetime and Stratospheric Influence. *47*(4), e2019GL084486. https://agupubs.onlinelibrary.wiley.com/doi/abs/10.1029/2019GL084486

Benedict, K. B., Zhou, Y., Sive, B. C., Prenni, A. J., Gebhart, K. A., Fischer, E. V., et al. (2019). Volatile organic compounds and ozone in Rocky Mountain National Park during FRAPPÉ. *Atmos. Chem. Phys.,* *19*(1), 499-521. https://acp.copernicus.org/articles/19/499/2019/

Goldberg, D. L., Vinciguerra, T. P., Hosley, K. M., Loughner, C. P., Canty, T. P., Salawitch, R. J., & Dickerson, R. R. (2015). Evidence for an increase in the ozone photochemical lifetime in the eastern United States using a regional air quality model. *Journal of Geophysical Research: Atmospheres,* *120*(24), 12778-12793. https://agupubs.onlinelibrary.wiley.com/doi/abs/10.1002/2015JD023930

Kong, H., Lin, J., Zhang, Y., Li, C., Xu, C., Shen, L., et al. (2022). Unexpected high NOX emissions from lakes on Tibetan Plateau under rapid warming. *Nature Geoscience (Accepted).*

Ma, J., Tang, J., Zhou, X., & Zhang, X. (2002). Estimates of the Chemical Budget for Ozone at

Waliguan Observatory. *Journal of Atmospheric Chemistry, 41*(1), 21-48.

http://dx.doi.org/10.1023/A%3A1013892308983

Mu, Y., Pang, X., Quan, J., & Zhang, X. (2007). Atmospheric carbonyl compounds in Chinese

background area: A remote mountain of the Qinghai-Tibetan Plateau. *112*(D22).

https://agupubs.onlinelibrary.wiley.com/doi/abs/10.1029/2006JD008211

Wang, T., Wong, H. L. A., Tang, J., Ding, A., Wu, W. S., & Zhang, X. C. (2006). On the origin

of surface ozone and reactive nitrogen observed at a remote mountain site in the

northeastern Qinghai-Tibetan Plateau, western China. *Journal of Geophysical

Research: Atmospheres, 111*(D8), D08303. http://dx.doi.org/10.1029/2005JD006527

Wu, F., Yu, Y., Sun, J., Zhang, J., Wang, J., Tang, G., & Wang, Y. (2016). Characteristics,

source apportionment and reactivity of ambient volatile organic compounds at Dinghu

Mountain in Guangdong Province, China. *Science of The Total Environment, 548-549*,

347-359. https://www.sciencedirect.com/science/article/pii/S0048969715310512

Xu, W., Xu, X., Lin, M., Lin, W., Tarasick, D., Tang, J., et al. (2018). Long-term trends of

surface ozone and its influencing factors at the Mt Waliguan GAW station, China –

Part 2: The roles of anthropogenic emissions and climate variability. *Atmos. Chem.

Phys., 18*(2), 773-798. https://www.atmos-chem-phys.net/18/773/2018/

---

## Author Comment (AC2)

**Response to Reviewer #2:**

**Reviewer #2:** This paper reports spring and summer season $O_3$ and PAN levels and their variations in Tibet and presents an analysis of what has been controlling the changes of those two most important photochemical air pollutants at such a high-altitude site under very clean conditions. Different impacts of transport and photochemistry on $O_3$ and PAN have been detected, revealing distinct transport pathways and different sensitivity to photochemical formation processes. Overall, the topic of the paper fits well into the scope of Atmospheric Chemistry and Physics, the observation and analysis methods used are scientifically sound and the results revealed sufficient novelty. The manuscript can be accepted for publishing after the following questions have been well addressed.

**Response:** We thank the reviewer for the detailed review of our manuscript, for all the valuable suggestions, and for approving the value of this research. In the following we provide point-to-point responses to the detailed comments and suggestions.

1. Although the authors reported the contributions of transport or STE and local chemistry to the observed $O_3$ and PAN concentrations, most of the results are descriptive, particularly before section 3.3. It's better to discuss the results more quantitatively and compare them with relevant studies.

**Response:** We thank the reviewer for this suggestion and added more quantitative discussions in Sect. 3.1 and 3.2, providing concentration levels of $O_3$, PAN and their precursors during spring and summer, making comparisons with measurements obtained at other background mountain sites. Detailed changes can be seen in the revised manuscript with tracked changes.

2. The writing style needs to be improved. It is difficult to understand some conclusions because some key information in the figures is omitted.

**Response:** Thanks for the suggestions, we went through the entire manuscript (including all figure and table captions) and improved its readability.

3. Almost all the modeled PAN concentrations are higher than the observed values, while it is completely opposite for $O_3$. My question is whether it is suitable to evaluate the importance of local chemistry for $O_3$ and PAN chemistry using a box model under an extreme environment such as Tibet, where is greatly affected by transport or STE?

**Response:** Thank you for posing this question. Since the box model was set to represent a surface air parcel, it can only account for near surface chemical processes. The F0AM model is based on the latest version of the most

detailed near-explicit chemical mechanism (Master Chemical Mechanism v3.2), with sufficient observation constraints on $O_3$ and PAN precursors, as well as sufficient spin-up time to warm up the model, the local photochemistry should be relatively well represented by the model. In the extreme environment of Tibet, STE affects $O_3$ much more than local photochemistry does, making it difficult to assess the influence of chemical process on $O_3$ and PAN formation, which is exactly why the measurement-constrained box model was adopted, in the hope to provide more insight into the influence of chemical formation on the variation of $O_3$. Simulated PAN was consistently higher than observed ones, especially when $NO_2$ was constrained, suggesting that under observed VOCs and $NO_2$ levels, local chemical formation would result in even higher PAN concentrations, if it were not for the impact of all kinds of transport processes. Overall, we believe it was suitable to this box model in the evaluation of local photochemistry (that was overshadowed by transport processes within observation data), however, this does not mean that the box-model can be deployed to predict $O_3$ and PAN at such locations, since transport process had too much impact on their overall concentration variations.

4. Did you warm up the model before simulating? How did you evaluate your model performance?

**Response:** Thanks for the question. We allowed the model to have a 3-day spin-up time, allowing the formation of various VOCs and intermediate oxidation products. We evaluated the model performance by comparing modelled $O_3$ net formation with those derived in another study of ours (Y.R. Wang et al., to be submitted to this special issue) based on photo stationary state assumptions using measurement constraints on $O_3$ precursors as well as radicals (OH, $HO_2$ and $RO_2$), which agreed very well with each other. Both studies concluded positive net $O_3$ formation. Additionally, we made sure that modelled NO concentrations fell into a reasonable range ($21\pm38$ ppt).

5. Figure 2b, it is difficult to differentiate the different components of VOCs. Figure 3b, what are the dotted lines

**Response:** Thanks for pointing that out, this was caused by the insufficient resolution of the figure within the preprint version. We modified the figure as the following:

[Figure]

**Figure 1**. Timeseries of a,g) $O_3$ (black), PAN (red), b,h) VOCs (bars), Y index (black), PBLH (blue), c,i) $NO_2$ (black), CO (red), d,j) $jO^1D$ (black), $jNO_2$ (red), e,k) RH (blue), T (red), f,j) wind

speed and wind direction (colored dots) during the spring (a-f, 1 May to 15 Jun.) and summer (g-j, 15 Jul. to 30 Jul.) period at Nam Co.

6. Please define the w wind, u wind, v wind.

**Response:** Thank you for the suggestion, we added definitions to the caption of Figure 4, where they first appeared.

7. Line 28: "However, only 0.1% of the springtime day-to-day $O_3$ variability **could by** STE processes, …". Something is missing in this sentence.

**Response:** Thank you for noticing, we must have deleted some words by accident. We corrected this sentence to:

"However, only 0.1% of the springtime day-to-day $O_3$ variability could be explained by STE processes, while 22% was explained during summertime."

8. Line 32: Make sure you really mean "diurnal" here and not "day-to-day".

**Response:** Thanks for pointing that out. We corrected this sentence to:

"Near surface photochemical formation was unable to account for the high $O_3$ level observed at NMC, and nor was it the determining factor for the day-to-day variability of $O_3$. However, it was able to capture events with elevated PAN concentrations and explain its day-to-day variations."

9. Line 115: There should be a comma before "which"

**Response:** Thanks, a comma was added.

10. Line 288-291: South Asian countries obviously strong pollution sources close to Tibet, revealing PAN transport, especially in summer, however, $O_3$ transport seemed weaker in comparison (Fig. 8). Is that because of enhanced dry deposition in southerly low altitude airmasses?

**Response:** There are several reasons behind the fact that southerly airmasses were not obviously linked to high $O_3$ concentrations. First, tropospheric $O_3$ concentrations typically increase with altitude, thus airmasses travelling over higher altitudes are more likely linked to higher $O_3$ concentrations. Secondly, airmasses travelling over heights near earth's surface are influenced by dry deposition as the reviewer suggested. Last, airmasses influenced by fresh stratospheric transport carry much higher $O_3$ loadings than tropospheric airmasses, which is why high $O_3$ were mostly linked to westerly and northerly airmasses during spring and summer, respectively.

From the fact that northerly low altitude airmasses that originated from the stratosphere revealed significantly elevated $O_3$, we might deduce that dry deposition might not have had such a strong influence on the depletion of $O_3$ during its transport.

11. Line 339: make sure of the consistency between "air mass" and "airmass".

**Response:** Thanks for the suggestion, we went through the manuscript and changed them all to air mass.

12. Lines 411-412: Since local photochemistry highly overestimates observed concentrations, does that mean that free tropospheric input and PBL growth results in diluted PAN concentrations?

**Response:** That is a very good point. In fact, prenoon PAN concentrations did not continuously grow with increasing PBLH, only revealing moderate levels at high PBLH. This suggests that intense free tropospheric input might have diluted near surface PAN concentrations. Additionally, the overestimation might also have resulted from the fact that we assumed observed PAN were formed at near surface level in the box model. However, since winds were very strong at Nam Co, observed daytime PAN concentrations might have been more affected by free tropospheric PAN formation, rather than surface formation.

13. Lines 420-421: OVOCs contributed overall largely to the total VOCs concentrations and the authors suggest that PAN revealed certain sensitivity towards OVOCs. Which OVOCs contributed most to the formation of PAN?

**Response:** We thank the reviewer for the question. According to the modelling results (Fig. S17 below), the production of PA radicals (that formed PAN upon combining with $NO_2$) was dominated by the oxidation of acetaldehyde, while the oxidation of methylglyoxal, biacetyl and peroxyacetic acid also made small contributions. This conclusion was added to Sect. 3.4:

*"Compared to $O_3$, PAN was sensitive to concentrations of both OVOCs and $NO_2$, since some of the OVOCs are direct precursors of PA radicals, which combine with $NO_2$ in PAN formation. **According to modelling results in Fig. S17, acetaldehyde oxidation contributed majorly to PA radical formation at Nam Co (71.8%), followed by methylglyoxal (9.0%) and biacetyl (5.1%).** Still, $NO_2$ was more decisive of the overall $O_3$ and PAN production, since without its constraint, $O_3$ net loss and negligible PAN net formation would be yielded."*

[Figure]

**Figure S17.** Average relative contribution of first-generation precursors to PA radical formation.

14. Line 445-446: "…, airmasses rich in $O_3$ were mainly associated with high altitude westerly airmasses that either entered the TP from the west **of** from the south, …" Replace "of" with "or"?

**Response:** Thanks for noticing, we made an according adjustment.